# Advanced Proteomic and Bioinformatic Tools for Predictive Analysis of Allergens in Novel Foods

**DOI:** 10.3390/biology12050714

**Published:** 2023-05-13

**Authors:** María López-Pedrouso, José M. Lorenzo, Juan de Dios Alché, Ramón Moreira, Daniel Franco

**Affiliations:** 1Department of Zoology, Genetics and Physical Anthropology, Universidade de Santiago de Compostela, Santiago de Compostela, 15872 A Coruña, Spain; mariadolores.lopez@usc.es; 2Centro Tecnolóxico da Carne de Galicia, Rúa Galicia Nº 4, Parque Tecnológico de Galicia, San Cibrao das Viñas, 32900 Ourense, Spain; jmlorenzo@ceteca.net; 3Plant Reproductive Biology and Advanced Microscopy Laboratory, Department of Biochemistry, Cell and Molecular Biology of Plants, Estación Experimental del Zaidín, Spanish National Research Council (CSIC), Profesor Albareda 1, 18008 Granada, Spain; juandedios.alche@eez.csic.es; 4Department of Chemical Engineering, Universidade de Santiago de Compostela, 15782 Santiago de Compostela, Spain; ramon.moreira@usc.es

**Keywords:** novel proteins, food safety, allergenicity, mass spectrometry, omic technologies

## Abstract

**Simple Summary:**

Food proteins from new sources such as vegetable origin (pulses, legumes, cereals), fungi, bacteria and insects are being introduced into the market. However, these novel foods, which had often not been consumed by humans, pose an important risk to public health. The biggest challenge is to ensure food safety by analysing in detail their compositional, nutritional, toxicological and allergenic properties. As a massive preliminary screening, proteomic methods should be employed to search for potential allergens. This review focuses on proteomic and bioinformatic tools for food researchers to identify allergens in novel foods. There is a multitude of highly valuable online tools and protein databases based on sequence alignment, motif identification or 3-D structure predictions. Thus, plant and animal food allergens, including lipid transfer proteins, profilins, seed storage proteins, lactoglobulins, caseins, tropomyosins, parvalbumins and other similar proteins, could be detected in novel food matrices. Furthermore, novel potential allergens could be found for further analysis. This would imply a major simplification.

**Abstract:**

In recent years, novel food is becoming an emerging trend increasingly more demanding in developed countries. Food proteins from vegetables (pulses, legumes, cereals), fungi, bacteria and insects are being researched to introduce them in meat alternatives, beverages, baked products and others. One of the most complex challenges for introducing novel foods on the market is to ensure food safety. New alimentary scenarios drive the detection of novel allergens that need to be identified and quantified with the aim of appropriate labelling. Allergenic reactions are mostly caused by proteins of great abundance in foods, most frequently of small molecular mass, glycosylated, water-soluble and with high stability to proteolysis. The most relevant plant and animal food allergens, such as lipid transfer proteins, profilins, seed storage proteins, lactoglobulins, caseins, tropomyosins and parvalbumins from fruits, vegetables, nuts, milk, eggs, shellfish and fish, have been investigated. New methods for massive screening in search of potential allergens must be developed, particularly concerning protein databases and other online tools. Moreover, several bioinformatic tools based on sequence alignment, motif identification or 3-D structure predictions should be implemented as well. Finally, targeted proteomics will become a powerful technology for the quantification of these hazardous proteins. The ultimate objective is to build an effective and resilient surveillance network with this cutting-edge technology.

## 1. Introduction

Food adverse reactions can mainly be classified as food allergies involving immune mechanisms. In developed countries, they are becoming more and more frequent, and the most common treatment available is to avoid allergen consumption [1]. Allergens are generally defined as any substances that can cause allergic reactions through the mediation of the immune system. Several substances from pollen, foods and others react with specific antibodies called Immunoglobulin E (IgE), leading to allergy symptoms in the patient. They can travel to cells, releasing chemicals and causing symptoms mainly on the nose, lungs, throat, sinuses, ears, lining of the stomach and skin [2]. Different immune mechanisms underlie food allergic reactions, mostly arising from IgE-mediated responses. There are many factors which determine the allergic immune response to allergens. During early childhood, IgE-associated food allergies are already present. Allergens induce IgE production in genetically predisposed individuals during primary sensitization. Particularly, the interleukins and other cytokines activated and secreted by the action of allergens induce IgE antibody production. The interaction of IgE and the target cells (mast cells, basophil granulocyte cells) leads to a body hypersensitive. Later, allergens can activate allergen-specific T cells and induce IgE responses in the secondary immune response [3,4]. Regarding food allergies, the development of regulatory T cells could be replaced by the generation of T helper 2 (Th2) cells, which leads to IgE class switching and the expansion of allergic effector cells [5]. Most of the allergenic reactions are caused by proteins that sometimes have some common characteristics, such as great abundance in foods, small molecular mass (<70 kDa), usually glycosylated, water-soluble and great resistance to proteolysis during cooking and digestion [6]. Epitopes are defined as the chemical groups on the surface of an allergen which specifically react with antibody or antigen receptors and are of the utmost importance for the allergy reaction. In the case of protein allergens, epitopes consist of a small sequence of amino acids located linearly and continuously in the primary structure (linear epitopes) or discontinuous but adjacent in a three-dimensional structure (conformational epitopes) [7]. For this reason, patients suffering from a specific allergy may experience allergic symptoms due to other allergenic proteins based on their similarity in chemical structure, a feature termed cross-reactivity. Thus, for instance, patients who are allergic to pollen could also have allergenic reactions after eating several fruits. Cross-reactivity usually occurs when the similarity between primary and secondary structure is greater than 50–70%, and then antigens from the second allergenic source, with similar three-dimensional structural regions (i.e., epitopes), are recognized by antibodies present against the primary source, thus triggering the allergy. The diagnosis of cross-reactivity can be challenging in food allergies. Even similar proteins could be detected by a positive skin test or blood test (serum IgE), but the allergic patient may not have any allergic symptoms from eating that food, even containing these proteins. However, as noted above, this review is focused on searching for potential allergens for subsequent clinical trials. Based on these facts, prospective identification of potential cross-reactivity and epitope mapping of likely allergens is becoming more crucial for developing new foodstuffs. In summary, the identification of allergens in novel foods could be based on the comparison of the protein structures to search for the allergenicity linear motifs or IgE-binding epitopes. Modelling the tertiary structure using computational methods could also lead to the identification of potential allergens for conformational epitopes of allergens.

## 2. Allergens of Traditional Food

It is well established that several proteins can induce an allergic response with fatal consequences in susceptible individuals. Symptoms from erythema to anaphylactic shock could be provoked by the interaction between the protein and the immune system in a complex way. For this reason, they are very difficult to predict, and the strategy generally followed by the food industry is to identify those proteins and the characteristics that induce allergic sensitization and allergic disease [8]. Most food allergies are caused by nine foods termed “the Big-9”, which are egg, fish, milk, peanut, shellfish, soy, tree nuts, wheat and sesame. Labelling of foods containing these allergens is mandatory according to the USA, Canadian, Japanese and Australian/New Zealand regulations. Additionally, the list of food ingredients declared as allergens in the EU rises to 14 foods: cereals containing gluten, crustaceans, eggs, fish, peanuts, soybeans, milk, nuts, celery, mustard, sesame seeds, sulphur dioxide and sulphites, lupin and molluscs. There is a broad consensus in that labelling, control and legislation should be referred to the specific allergenic molecules and their bioavailability for better food safety [9,10,11]. As shown in Table 1, the subcommittee of the World Health Organization/International Union of Immunological Societies (WHO/IUIS) recommended the nomenclature of allergens consisting of the first three letters of the genus and the first letter from the species source of the allergen, followed by sequential Arabic numerals indicating the order of its description. For instance, Ara h 1 allergen from peanut (*Arachis hypogaea*) is a cupin (7S globulin) which is an important seed storage protein located in the cotyledons. Although food allergens are difficult to categorize due to the huge variety, and food labelling policies of allergens should be improved and standardized to ensure the safety of allergic. Furthermore, once a candidate allergen is detected by researchers, the information should be submitted to Allergen Nomenclature Sub-Committee, and an evaluation process must be conducted for incorporation of the new allergen in this database. This fact highlighted the importance of obtaining an official allergen nomenclature. In recent years, allergen identifications and the establishment of databases providing molecular, structural and clinical data should be increased. Cow milk allergens could be divided into two main groups of proteins caseins (αS1-casein, αS2-casein, β-casein, and κ-casein), which precipitate at pH 4.6 and 20 °C, and soluble proteins of the serum-like β-lactoglobulin, α-lactalbumin, bovine lactoferrin, bovine serum albumin and bovine immunoglobulins. Nevertheless, it is clear that major allergens are caseins, β-lactoglobulin and α-lactalbumin [12]. The six major allergens of the egg are ovomucoid, ovalbumin, ovotransferrin, and lysozyme from egg white and α-livetin and YGP42 from the yolk. However, egg allergy occurs mainly due to the proteins in the egg white [13]. In the case of fish, the major allergen is parvalbumin, mainly β-parvalbumin, as well as other lesser-known proteins. The safety of fish consumption becomes worse because of various toxins and parasites, including ciguatera and Anisakis [14]. Parvalbumin and tropomyosin are also allergens in shellfish. It was demonstrated that tropomyosin causes a high cross-reactivity in crustaceans, insects and other molluscs [15]. Peanuts and tree nuts pose a safety risk because they could trigger fatal anaphylaxis even in small amounts. A total of 16 allergens were officially included by WHO/IUIS Allergen Nomenclature Sub-Committee, which can be classified into 7 groups [16]. There are also soybean-allergic individuals, and a total of 15 proteins were identified as allergens in soy hydrolysates. Among them, β-conglycinin and glycinin are the most studied by their great abundance [17]. Regarding allergens from wheat, the most important are inhibitors of α-amylase and trypsin, as well as α/β-, γ- and ω-gliadins. To a lesser extent, LMW-glutenins, lectins (WGA), and possibly also lipid transfer proteins are identified as allergens [18]. Finally, sesame has been considered a source of allergens very similar to other vegetal seeds [19]. As may be seen, allergens in traditional foods are quite safe, but cutting-edge products should receive more attention.

## 3. Allergens of Novel Foods

Greater environmental awareness and increasing health concerns will be the major trends in the years to come. Concerning the first point, the high impact of agro-industrial activities on the environment, climate change and animal welfare is encouraging food scientists to search for alternative protein sources. Secondly, consumer demands healthier products, including bioactive peptides derived from novel protein hydrolysis. Antioxidant, antihypertensive, hypocholesterolemic and anticancer activities among others, are being sought in a wide range of protein sources. In this regard, novel foods are becoming more relevant, contributing to enhancing several of these aspects. Vegetables, insects, and microorganisms could meet the nutritional protein quality, but sensorial aspects, neophobia and those related to food regulation have not yet been resolved. However, potential hazards for these novel foods, including contaminants (heavy metals, mycotoxins, pesticide residues), new pathogens and allergens, have to be overcome [20]. The most important challenge concerning novel foods is ensuring food safety. Within this framework, we will focus on the allergens of novel foods. Allergens are highly heterogeneous molecules from both the animal and the vegetable kingdoms, as shown in Figure 1.

Regarding vegetal families of allergens, 2 S albumins, non-specific Lipid Transfer Proteins (nsLTP), cereal α-Amylase Trypsin Inhibitors (ATI) and cereal prolamins, legumins and vicilins (cupin superfamily), profilins, and Pathogenesis-Related (PR)-10 proteins are the main groups of vegetal allergens [21]. As a practical approach to studying allergens in novel vegetal foods such as seeds, a strategy to identify the allergens in foods is to use the antibody cross-reactivity between storage proteins. Thus, cross-reactivity of certain antibodies from sera of sesame hazelnut and peanut-allergic patients was assessed in chia seed resulting in a high similarity of epitopes on globulins of chia seed and sesame [22]. On the other hand, the identification of peptide markers of these proteins is very useful in other tree nuts (e.g., hazelnut, chestnut, pecan and walnut). However, this strategy has major drawbacks, such as molecular heterogeneity of proteins, including isoforms and other differences as well as limited annotated protein sequences in databases [23].

In the case of animal foods, the most important allergens are tropomyosins, the EF-hand family (parvalbumins), the ATP: guanido phosphotransferases (arginine kinases) and the α/β-caseins [24] from fish, shellfish, and milk. However, other novel foods may also include less-studied food safety issues. In the case of edible insects, they have an attractive nutritional profile and lower feed conversion ratio. The most often consumed are mealworms (*Tenebrio molitor*), house crickets (*Acheta domesticus*) and lesser mealworms (*Alphitobius diaperinus*) larvae and all were related to allergenic reactions. It has been demonstrated that tropomyosin and arginine kinase are the most common allergens from insects. The allergenicity could be studied from the cross-reactivity and/or co-sensitization with tropomyosin and arginine kinase of house dust mites and seafood (usually prawn and shrimp) [25,26]. Indeed, it has been demonstrated that the majority of shrimp-allergic patients are at risk for mealworm allergy. This allergy could be caused by major shellfish allergens: tropomyosin and arginine kinase, as well as other minor allergens: sarcoplasmic calcium-binding protein and myosin light chain [27]. However, contradictory findings were reported by Francis et al. (2019), who determined limited cross-reactivity of arginine kinase from mealworm and cricket insects [28]. Additionally, a large number of putative allergenic as aldolase, α-amylase, aspartic protease, chitinase, cockroach allergen group 1, cysteine protease, glutathione-S-transferase, heat shock protein 70, hemocyanin/hexamerin, myosin heavy and light chains, serine protease (trypsin), triosephosphate isomerase and troponin C were identified in *Tenebrio molitor*, and they are also official insect allergens from the WHO/IUIS systematic allergen nomenclature [29]. Nevertheless, it has been demonstrated that boiling, frying and roasting greatly reduce the safety risk of edible insects [30]. For instance, cross-reactivity and allergenicity in *Locusta migratoria* after food processing, such as extraction methods, enzymatic hydrolysis and thermal treatments, could be deleted [31]. The use of alcalase for enzymatic hydrolysis in cricket (*Gryllodes sigillatus)* also produced a decrease in IgE reactivity to tropomyosin [32]. Table 2 is summarized some examples of allergens from novel foods.

A microbial protein, referred to as a single-cell protein, is another relevant source of protein produced by microalgae, fungi, yeast, or bacteria. Scarce information is available about their food allergies, and they appear to be restricted. However, the safety risk is more associated with pathogens, toxins and contaminants (heavy metals, hydrocarbons, etc) [33]. Microalgae, tablets/capsules, snacks, pasta, cookies, bread and so on are elaborated from spirulina (*Arthrospira platensis*) and chlorella (*Chlorella vulgaris*). Several authors have reported allergic reactions after the consumption of microalgae products. Anaphylaxis is caused by the consumption of spirulina-based products [34,35], and acute tubulointerstitial nephritis follows the ingestion of chlorella tablets [36]. However, the risk assessment has not been intensively studied, including allergenic reactions. An allergenic protein called β-chain of phycocyanin C from spirulina protein extracts has been reported [35]. In a more recent article, several putative allergens were found in spirulina and chlorella after a proteomic analysis and in silico sequence homology prediction [37].

**Table 2 biology-12-00714-t002:** Presence of allergens in novel foods based on microalgae and insects.

Novel Food	Protein Name/Allergen	Specie	Refs.
Microalgae	C-phycocyanin Thioredoxins Superoxide dismutase Glyceraldehyde-3-phosphate dehydrogenase Triosephosphate isomerase	*Microalgae spirulina (A. platensis)*	[35,37]
Microalgae	viz. calmodulin Fructose-bisphosphate aldolase	*Microalgae chlorella (C. vulgaris)*	[37]
Insects	Tropomyosin, myosin, actin, troponin C (muscle proteins) Tubulin (cellular proteins) Hemocyanin, defensin (circulating proteins) Arginine kinase, glyceraldehyde 3-phosphate dehydrogenase (GAPDH), triosephosphate isomerase, α-amylase, trypsin, phospholipase A, hyaluronidase (enzymes)		[38,39]

Additionally, the allergenicity issue is altered by changes in protein modifications during the cooking process, digestion and others. In some cases, the application of innovative thermal and non-thermal processing of food has a great impact on several allergens, modifying food immunoreactivity [40,41]. For instance, many processing procedures, including steam boiling, microwave heating, and enzyme or ethylene treatments, reduce banana and kiwifruit allergenicity [42,43]. For these reasons, all the technologies related to the identification of proteins and structural knowledge about their post-translational modifications and interactions within the food matrix help to control allergenicity.

## 4. Current Prevalent Methods Used to Assess the Presence of Allergens in Foods

Currently, the identification of food allergens is widely addressed by antibody-based assays for the direct measurement of IgE-binding molecules. In food research, the Enzyme-Linked ImmunoSorbent Assay (ELISA) is the most widely used method for allergen detection and quantification. Moreover, SDS-PAGE protein profiling of food extracts followed by Western immunoblotting with sera from allergenic patients is commonly used [44]. These methods also present several drawbacks, including false positives and false negatives due to the interaction of the antibodies with matrix components and sometimes the limited capacity to detect cross-reactivity phenomena. As an example, three commercially available ELISA kits were compared to detect lupine allergens and cross-reactivity with similar legumes, resulting in a wide variation in the calculated concentrations [45]. In another case, commercial fish ELISA kits were used to detect bony and cartilaginous fish in different foodstuffs. The quantification of these food allergens was unreliable, and the results depended on fish species and food matrix [46]. Therefore, more reliable, accurate and reproducible methods are needed to reduce the risk of allergic reactions in consumers. The main topic is to highlight this new proteomic and bioinformatic approach that could help to address the detection and labelling of allergens in novel food. Nowadays, it is increasingly used to screen potential allergens in novel foods. Among these studies, there are numerous examples of searching for potential allergens in novel foods such as Moringa oleifera leaves [47], silkworm pupa [48], plant-derived food [49], Chlamys nobilis [50] and others. Consequently, proteomic and bioinformatics are largely recognized as important tools in the analysis of allergens as well as the investigation of protein structural modifications produced by an industrial process which are so relevant in terms of food quality and safety. There are enough studies to prove that this strategy is fundamental for high-throughput screening of putative allergens. Furthermore, after a deep knowledge of food allergens, numerous industrial applications of proteins could be introduced, such as allergen biosensors [51]. A biosensor is a highly advanced technological device which consists of an integrated receptor–transducer unit. Its primary function is to convert a biological signal of recognition into a quantitative chemical or physical signal. For instance, the receptor of a biosensor could be an antibody that detects an allergen or a single-stranded DNA molecule that hybridizes with an allergen. This ability to accurately detect and even quantify biological signals is particularly relevant in the development of new monitoring tools for improving food safety [52]. For this reason, the detection and identification of allergens are of utmost importance. Biosensors offer a highly sensitive and specific method which could be especially relevant in the food industry and healthcare.

## 5. Proteomic Approach to Identify Allergens in Novel Foods

The implementation of mass spectrometry-focused proteomic methods in the field of food science, both through targeted and untargeted approaches, is currently increasing and is expected to be the most widespread technology in the field of food allergens in the next few years. Qualitative and quantitative proteomic analyses are usually carried out in two steps, including liquid chromatographic separation followed by mass spectrometry identification, and they are particularly relevant in complex and processed foods. There is a great concern about the risk of allergens in the consumption of novel protein sources. The food industry needs to include novel food matrices such as insects, seaweeds, microalgae, or non-common seeds to guarantee its sustainability as well as maintain food safety on global terms. The balance between human health risks and these challenges should be tackled by the food industry in the next few years. A brief bibliometric search collected from the Scopus database using the keywords “proteomic”, “allergen” and “food” was used to identify relevant documents published since 2020 until now (Table 3).

The search showed that the most common food matrix studied in this sense was the novel food (insects, seaweeds and rare vegetables) to evaluate new potential allergens. New and modified proteins may present a de novo sensitization risk, as demonstrated in studies with mealworms. However, the most common strategy focuses on cross reactivity resulting in overestimation or underestimation of allergic risk [65]. Regarding the bioinformatic analysis, the most frequently used tools are listed and detailed in Table 4. Identifying and quantifying all allergens in each food matrix is of the utmost importance for systematic analyses. The recent efforts in allergen data collection represent a great challenge, and bioinformatic tools need to be further developed to be useful.

Shotgun proteomics is used to study complex mixtures of proteins through the detection of specific peptides generated after proteolysis by trypsin [75,76]. The whole proteome could be analysed to search for new potential allergens using in silico bioinformatic tools. In this sense, online databases such as UniProt and Basic Local Alignment Search Tool (BLAST) make it possible to predict allergenicity. A very relevant database for identifying protein allergens is the UniProt Knowledgebase linked to Allergome, which combines 1303 reviewed UniProtKB/Swiss-Prot and 3117 unreviewed UniProtKB/TrEMBL entries [70]. On the other hand, BLAST compares protein sequences searching for regions of similarity, being particularly useful in the case of cross reactivity. Other bioinformatic tools based on sequence alignment and motif identification can be used to predict allergenicity. The Food and Agriculture Organization (FAO/WHO) and the European Food Safety Authority (EFSA) recommended the potential IgE cross-reactivity based on an identity >35% over 80 amino acid sequences. However, experimental evidence has demonstrated that cross reactivity requires more than 70% identity in most situations [77]. Therefore, extensive knowledge of the proteins is necessary to predict the allergic response to novel food ingredients and additives based on homology [78]. Additionally, predictions of 3D structure from amino acid sequences can be modelled by using platforms such as AlphaFold Protein Structure Database to study and compare conformational epitopes [79].

## 6. Targeted Proteomics for Quantification of Food Allergens

Mass spectrometry (MS)-based targeted proteomics is a very suitable method to quantify target peptides chosen from the allergen protein sequence. In this regard, the Selected Reaction Monitoring mode (SRM), also known as Multiple Reaction Monitoring (MRM), can monitor the peptide marker from a precursor protein allergen with high specificity and sensitivity. In a previous step, the peptide marker should be carefully chosen, aiming to be allergen-specific and stable under processing conditions. Additionally, the quantification is performed through the standard addition of isotopically labelled peptide standards. For instance, three peach allergens were detected in a concentration of 0.4–2000 nmol/L with recovery yields higher than 95% in peach juice, peach can, jam, dried peach slice and peach yoghurt [80]. In another study, six soy-derived ingredients were assayed in different food matrices for allergen quantification, internal quality control and interlaboratory calibrations. The results demonstrated that MS methods had a higher capacity to detect and quantify highly processed soy proteins than ELISA kits [81]. In another study, peptides from α-S1-casein and β-lactoglobulin of milk were selected for searching caseinates in sausages, hamburgers and pâté samples with great success [82]. As an essential element in developing novel food products, bioinformatic tools are of paramount importance in finding new potential allergens for further validation. Table 4 summarizes basic bioinformatic tools for evaluating the potential allergens.

## 7. Conclusions

Proteomic approaches using advanced mass spectrometry will continue providing even more relevant information in the field of food safety (Figure 2). Detection, identification and quantification of known allergens in complex matrices and highly processed food have already been developed, and targeted mass spectrometry allows monitoring of them during food processing. However, the identification of novel protein allergens in insects, seaweeds, microalgae or other non-common vegetable foods is one of the most important challenges over the next few years. In this sense, bioinformatic tools and curated databases of allergens will enable the prediction of potential allergens, and these newly discovered allergens should be validated subsequently. This information could be used to improve the design and safety of food products by novel devices. Advanced technologies, including biosensors, could identify specific interactions between receptors and allergens, enabling us to address the challenges of food safety monitoring [83]. It is still necessary to make a great effort in this field.

## Figures and Tables

**Figure 1 biology-12-00714-f001:**
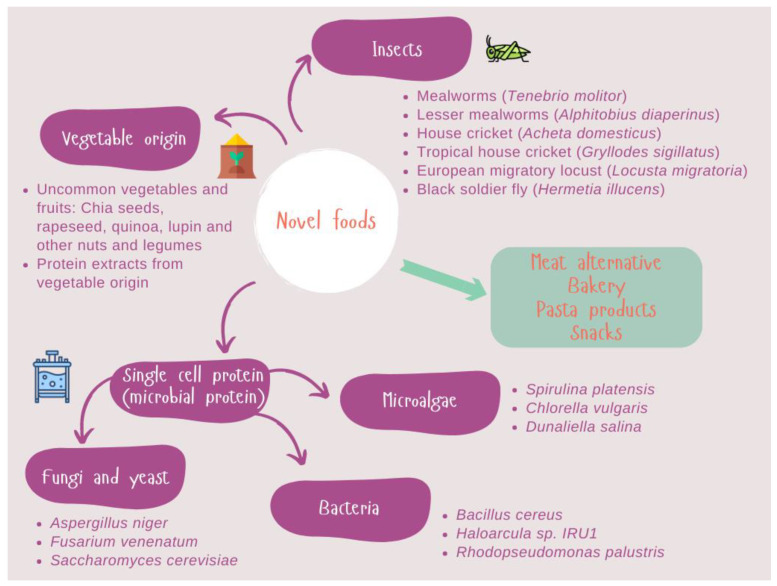
Novel protein sources of plant, algal, fungal and insect origins are being researched by the food industry.

**Figure 2 biology-12-00714-f002:**
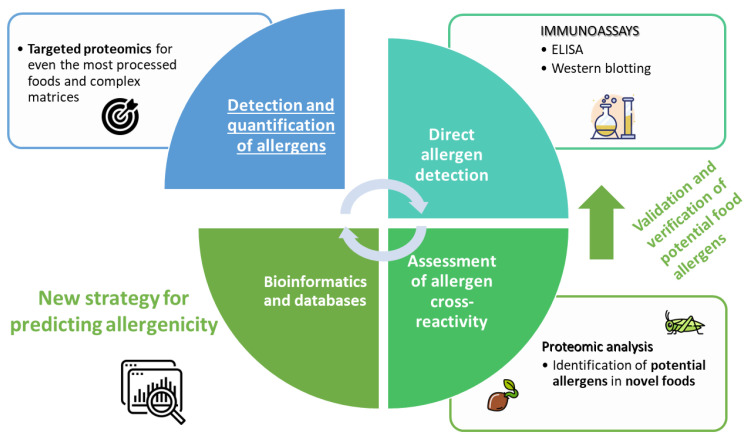
Schematic diagram of main trends to address the introduction of novel foods concerning their allergenicity.

**Table 1 biology-12-00714-t001:** Several protein allergens in food. For the WHO/IUIS nomenclature, the allergens are named according to the species source of food.

Food	Protein Name	Specie	Allergen	Ref.
Milk	Caseins α S1-casein (23.6 kDa) α S2-casein (25.2 kDa) β -casein (24 kDa) κ-casein (19 kDa) β-lactoglobulin (18.3 kDa) α-lactalbumin (14.2 kDa) Serum albumin (66.3 kDa) Immunoglobulin (160 kDa)	*Bos taurus*	Bos d 9 Bos d 10 Bos d 11 Bos d 12 Bos d 5 Bos d 4 Bos d 6 Bos d 7	[12]
Eggs	Ovomucoid (28 kDa) Ovalbumin (44 kDa) Ovotransferrin (78 kDa) Lysozyme (14 kDa) α-livetin (69 kDa) YGP42 (35 kDa)	*Gallus domesticus*	Gal d 1 Gal d 2 Gal d 3 Gal d 3 Gal d 5 Gal d 6	[13]
Fish	Parvalbumin α-parvalbumin (13 kDa) β-parvalbumin (11.6 kDa)	*Gadius callarias* (Baltic cod)	Gad p 2 Gad p 1	[14]
Shellfish	Tropomyosin (34 kDa)	*Metapenaeus ensis* (Shrimp)	Met e 1	[15]
Peanuts/tree nuts	7 S seed storage globulin, vicilins (64 kDa) 2 S albumin (17 kDa) Nonspecific lipid transfer proteins Oleosins Defensins Profilins Plant pathogenesis-related proteins PR-10	*Arachis hypogaea*	Ara h 1 Ara h 2, Ara h 6, Ara h 7 Ara h 9, Ara h 16, Ara h 17 Ara h 10, Ara h 11, Ara h 14, Ara h 15 Ara h 12, Ara h 13 Ara h 5 Ara h 8	[16]
Soy	7 S seed storage globulin, β-conglycinin 11 S seed storage globulin, glycinin	*Glycine max*	Gly m 5 Gly m 6	[17]
Wheat	α-amylase inhibitor (13 kDa) Gamma gliadin (88 kDa) Elongation factor 1	*Triticum aestivum*	Tri a 28 Tri a 20 Tri a 45	[18]
Sesame	2 S albumins 7 S vicilin-type globulin (45 kDa) Oleosins 11 S globulin, legumins Profilin	*Sesamum indicum*	Ses i 1, Ses i 2 Ses i 3. Ses i 4, Ses i 5 Ses i 6, Ses i 7 Ses i 8	[19]

**Table 3 biology-12-00714-t003:** Relevant examples of the recent use of proteomic approaches for the detection of allergens in novel foods.

Novel Food	Bioinformatic Tool	Goal/Main Achievements	Ref.
Vegetables			
Bread wheat spelt and rye	Database of Allergen Families-AllFam AllergenOnline Allergome	Comparison of allergenicity in cereal products	[53]
Cashews	BLASTP Search against AllergenOnline sequence	Analysis of allergen stability under heat treatment	[54]
Goji berries	AlgPred software hybrid approach	Identification of 11 IgE-binding proteins	[55]
Macadamia nut	AllergenOnline Immune Epitope Database Analysis Resource (IEDB)	Analysis of homology and linear epitope similarities to known allergens	[56]
Medicago sativa	COMPARE allergen database	Identification of three allergenic protein families	[57]
Lentil (Lens culinaris)	Blast2GO—Functional Annotation and Genomics	Quantification of major allergen proteins	[58]
White- and red-fleshed pitaya seeds	AllermatchTM webtool Algpred 2.0 AllerCatPro web server	Identification of five potential allergens	[59]
Seaweeds			
Spirulina and chlorella microalgae	AllergenOnline	Six proteins exhibit significant homology with food allergens	[37]
Insects			
Black soldier fly, yellow mealworm, lesser mealworm, house cricket and Morio Worms	Allergen nomenclature (WHO/IUIS)	Detection of arginine kinase and tropomyosin	[60]
Cricket	Allermatch TM webtool AlgPred 2.0 ABCPred Bepipred	Description of the impact of processing on allergenic reactivity of insect proteins.	[61]
*Cricket Acheta domesticus*	Database of Allergen Families-AllFam Allergen nomenclatura (WHO/IUIS) CLC Genomics Workbench 20.0.4. AllerCatPro web server	Identification of 20 putative allergens	[62]
Lesser mealworms, black soldier flies and their protein hydrolysate	AllermatchTM webtool	Identification of potential allergens by similarity to known allergens	[63]
Parasites			
*Anisakis simplex, Pseudoterranova decipiens,* and *Contracaecum osculatum*	Blast2GO—Functional Annotation and Genomics AllergenOnline AllerTOP web server ver. 2.0 PREAL web server	Prediction of 53 probable allergens in three species	[64]

**Table 4 biology-12-00714-t004:** Bioinformatic software tools most used for allergen analysis.

Name	Link (Website)	Description	Ref.
Allergen nomenclature	http://www.allergen.org (accessed on 12 February 2023)	Official site for the systematic allergen nomenclature provided by the World Health Organization and International Union of Immunological Societies (WHO/IUIS)	[66]
AllerBase	http://bioinfo.unipune.ac.in/AllerBase/Home.html (accessed on 12 February 2023)	Database of allergens detected as IgE-binding epitopes, IgE antibodies and cross reactivity. Allergen data such as experimental information on its allergenic activity and food source is compiled, resulting in a curated database.	[67]
AllerCatPro	https://allercatpro.bii.a-star.edu.sg/ (accessed on 12 February 2023)	Provides protein allergenicity potential prediction based on the similarity of amino acid sequence and 3D protein structure	[68]
AllergenOnline	http://www.allergenonline.org (accessed on 12 February 2023)	Provides sequence database of allergens to identify proteins and assess the potential risk of allergenic cross-reactivity. This database offers 2233 peer-reviewed sequences from 912 taxonomic protein groups (February 2021)	[69]
Allergome	http://www.allergome.org (accessed on 12 February 2023)	A website with detailed information on Allergenic Molecules (Allergens) causing an IgE-mediated (allergic, atopic) disease (anaphylaxis, asthma, atopic dermatitis, conjunctivitis, rhinitis, urticaria).	[70]
Comprehensive protein allergen resource (COMPARE allergen database)	https://comparedatabase.org/ (accessed on 12 February 2023	A database comprised of protein sequences of known allergens	[71]
Database of Allergen Families-AllFam	http://www.meduniwien.ac.at/allfam/ (accessed on 12 February 2023)	Comprises a resource for classifying allergens into protein families as well as biochemical properties and allergology significance	[72]
Immune Epitope Database and analysis resource (IEDB)	https://www.iedb.org (accessed on 12 February 2023)	Provides experimental data on antibody and T-cell epitopes to identify allergens and to assist in the prediction and analysis of allergenicity	[73]
Structural Database of Allergenic Proteins (SDAP)	https://fermi.utmb.edu (accessed on 12 February 2023)	Tool for testing the FAO/WHO allergenicity rules in new proteins and investigating cross reactivity, also offering information about protein sequence and structure	[74]

## Data Availability

Not applicable.

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
