# Peer review of "Advanced Proteomic and Bioinformatic Tools for Predictive Analysis of Allergens in Novel Foods"

_biology, 2023, doi:10.3390/biology12050714_

Round 1

Reviewer 1 Report

This work is about advanced proteomics and bioinformatic tools for predictive analysis of allergens in novel foods.

The topic is relevant in the field of food analysis, including besides the traditional food allergens, the allergens in novel food, since agro-industrial activities on the environment enhance scientists’ need to search for alternative protein sources. Furthermore, the methods for the detection of allergens in those food are described. A short description about biosensors should be included.

The manuscript is well written and descriptive. In the section 4 about current prevalent methods used to assess the presence of allergens in foods there are also methods based on mass spectrometry that can detect allergens. Furthermore, it could be useful to add some information about biosensors that are gaining ground to food analysis sector and more specifically to allergens detection.

The conclusions are consistent with the evidence and the references are appropriate.

The figures and tables are well described with respect with the information provided in this manuscript.

Author Response

Reviewer 1

Comments and Suggestions for Authors

This work is about advanced proteomics and bioinformatic tools for predictive analysis of allergens in novel foods.

The topic is relevant in the field of food analysis, including besides the traditional food allergens, the allergens in novel food, since agro-industrial activities on the environment enhance scientists’ need to search for alternative protein sources. Furthermore, the methods for the detection of allergens in those food are described. A short description about biosensors should be included.

The manuscript is well written and descriptive. In the section 4 about current prevalent methods used to assess the presence of allergens in foods there are also methods based on mass spectrometry that can detect allergens. Furthermore, it could be useful to add some information about biosensors that are gaining ground to food analysis sector and more specifically to allergens detection.

The conclusions are consistent with the evidence and the references are appropriate.

The figures and tables are well described with respect with the information provided in this manuscript.

Response: Dear reviewer, thanks for your nice word about the manuscript. Thanks for the point about bionsesors. In total agreement with you we think that biosensors are a rapidly evolving field of research and incorporating the information and references are crucial for ensuring the relevance of the work.

Therefore we have introduced additional information (lines 205-212 and lines 270-272) and two references (52 and 83). Both, help to support our arguments and provide the reader’s further.

Reviewer 2 Report

1. Please draw the structure of various foods.

2. Please provide more figures and tables for readers to understand.

3. Please combine some typical figures to discuss the relevant activity mechanism.

4. Some references should be added, such as:

Ultrasonics Sonochemistry, 2023, 93, 106295.

Chemical and Biological Technologies in Agriculture, 2023, 10(1),15.

There are grammatical errors.

Author Response

Reviewer 2

Comments and Suggestions for Authors

  1. Please draw the structure of various foods.

Response: Dear reviewer, we don’t understand this suggest, this don’t have relation with our review

  1. Please provide more figures and tables for readers to understand.

Response: Dear reviewer, In our opinion the review is well supported by figures and tables provided

  1. Please combine some typical figures to discuss the relevant activity mechanism.

Response: What mechanism? the Allergenicity has no specific mechanism. This is enough complex, and this is out of review scope

  1. Some references should be added, such as:

Ultrasonics Sonochemistry, 2023, 93, 106295.

Chemical and Biological Technologies in Agriculture, 2023, 10(1),15.

Response: Both references are very high level and interesting, but they have no relation with the review there are focused on the extraction of polysaccharides from Ginkgo biloba leaves and coconut peel using ultrasound. In both papers, there are no statements or references related to allergen or allergenicity. Our review address with allergenicity issues. Specifically, our review is focused on novel foods and analytical aspects related to assessing new allergens, but not on medical aspects, despite nowadays they are no doubt about the influence on food intakes and allergen diseases. We think this emphasis is important to understand the idea of the review.

Round 2

Reviewer 2 Report

Accept

Moderate editing of English language.